

# The Effects of Land Use on Soil Carbon Stocks in the UK

Peter Levy[1], Laura Bentley[2], Bridget Emmett[2], Angus Garbutt[2], Aidan Keith[3], Inma Lebron[2], and David Robinson[2]

[1]Centre for Ecology and Hydrology, Bush Estate, Penicuik, Midlothian, EH26 0QB, UK
[2]Centre for Ecology and Hydrology, Environment Centre Wales, Deiniol Road, Bangor, Gwynedd, LL57 2UW, UK
[3]Centre for Ecology and Hydrology, Lancaster Environment Centre, Library Avenue, Bailrigg, Lancaster, LA1 4AP, UK

**Correspondence:** Peter Levy (plevy@ceh.ac.uk)

**Abstract.** Greenhouse gas stabilisation in the atmosphere is one of the most pressing challenges of this century. Sequestering carbon in the soil by changing land use and management is increasingly proposed as part of climate mitigation strategies, but our understanding of this is limited in quantitative terms. Here we collate a substantial national and regional data set (15790 soil cores), and analyse it in an advanced statistical modelling framework. This produced new estimates of the effects of land use
on soil carbon stocks in the UK, different in magnitude and ranking order from the previous best estimates. Soil carbon stocks were highest in woodlands, followed by rough grazing and semi-natural grasslands, then improved grasslands, and lowest in croplands. Estimates were smaller than the previous estimates, partly because of new data, but mainly because the effect is more reliably characterised using a logarithmic transformation of the data. With the very large data set analysed here, the uncertainty in the differences among land uses was small enough to identify consistent mean effects. However, the variability in
these effects was large, and this was similar across all surveys. This has important implications for agri-environment schemes, seeking to sequester carbon in the soil by altering land use, because the effect of a given intervention is very hard to verify. We examined the validity of the "space-for-time" substitution, and although the results were not unequivocal, we estimated that the effects are likely to be over-estimated by 5-33%, depending upon land use.

## 1   Introduction

Conversion of land to agricultural use has a had a significant impact on the global carbon cycle (Le Quéré et al., 2009, Wilken-skjeld et al. (2014), Obermeier et al. (2021)). This is expected to continue with the rising human population and demand for agricultural land in some regions, and moves towards reforestation and rewilding in other regions (Gitz and Ciais, 2004, Levy et al. (2004), Lawrence et al. (2016)). Different land uses affect the inputs of organic matter to the soil (via plant litter, crop residues, root death, and animal necromass), and affect the losses via heterotrophic respiration and leaching. Depending on
how a given land use affects the balance of these gains and losses, the soil carbon stock ($S_c$) may be increased or decreased, relative to its previous state (Ostle et al., 2009). Under the UNFCCC agreement, each nation is required to estimate its net emission or sequestration of carbon attributable to land-use change, as accounted for under the LULUCF reporting rules (Intergovernmental Panel on Climate Change (IPCC), 2003). The default method involves estimating the difference in the long-term equilibrium soil carbon stock among different land uses. This is then used, along with data on the area of land undergoing the





transition from each land-use type to every other land-use type (i.e. a matrix of land-use change), to dynamically model the change in soil carbon for each land-use transition each year. The key parameters in this model are the equilibrium soil carbon stock values for the different land uses. In the UK, these are estimated using a space-for-time substitute approach, based on a nationwide survey of soil carbon stock, stratified by land use and soil series (Milne and Brown, 1997, Bradley et al. (2005)). By examining the differences in soil carbon stock among land uses occurring on the same soil series, and averaging across

>400 soil series which cover the UK, we can estimate the overall mean effect of each land use on soil carbon stock.

Data from several other soil surveys have become available since 2005, but these differ in their methods, most critically in terms of the soil depth sampled, with many focussing on the more easily-sampled top soil. In the absence of measurements over the whole profile, these data have not previously been used to update the estimates of UK soil carbon stocks of Bradley et al. (2005), or the effects of land-use change. However, there are some issues with the approach of Bradley et al. (2005)

which makes updating these estimates timely. Bulk density was generally estimated using pedotransfer functions, which use organic carbon content as a predictor, and quite how this non-independence may affect the results is unclear. Some of the details of the analysis with respect to land use are not completely clear, and different choices and assumptions (about the land-use classification or data transformations, for example) can give quite different results.

A weakness to the general approach is in the assumption that the space-for-time substitution is fully valid: we assume that

the spatial differences observed in soil carbon stock within a soil series are completely attributable to differences in land use. This will be inappropriate if there are systematic pre-existing differences in the soil carbon stock on the land chosen for particular uses (such as arable crops, pasture, forestry). For example, the flat, low-lying, free-draining soils typically chosen for arable crops might differ substantially from the higher elevation, marginal land typically chosen for forestry. If there are such differences, this will confound the analysis, such that we are wrongly ascribing cause and effect.

The alternative to the space-for-time substitution approach is to measure the effect of a land-use change at a single site repeatedly over time. However, the number of such studies is very few because of the long time scale required to detect change and the large sample size needed to detect change above the local-scale variability (Schrumpf et al., 2011). Paired-plot or chronosequence studies represent a compromise, whereby the space-for-time substitution assumption is still made, but the spatial variability is minimsed by choosing closely co-located plots. However, the variability found in such studies is still rather

large (Poeplau et al., 2011). Some survey data have suggested that change in soil carbon has occurred over time, irrespective of land-use change (Bellamy et al., 2005, Reynolds et al. (2013)), though whether changes are real, general, or what their cause is is subject to debate (Smith et al., 2007, Chamberlain et al. (2010)). Although most of the survey data cover only a single short period of time, the Countryside Survey data are repeated measurements within the same 1-km squares since 1978. Although bulk density measurements were missing initially, these can still be used to examine possible trends in time (Thomas et al.,

55    2020).

The work described here attempted to improve quantification of the effects of land use on soil carbon stocks in the UK by (i) incorporating more recent survey data, (ii) applying more sophisticated statistical modelling, and (iii) rigorously assessing the validity of the assumption behind the space-for-time substitution. As a check for confounding variation, we look for evidence of change over time not caused by land-use change where the data allow this.





## 1.1 Hierarchical model

Different soil surveys have been carried out for different purposes, using different protocols, and differ in the depth that is sampled. For example, the CEH Countryside Survey (Emmett, 2010) has a large spatial coverage, with repeated sampling at the same locations, but only samples the top 15 cm of soil. Other surveys have sampled down to a depth of 1 m or more, but have limited coverage of land uses and only occur at one point in time (e.g. Ward et al., 2016). Measurements of bulk density, essential for calculating $S_c$ via Equation 4, are sometimes missing and may have to be imputed from pedotransfer functions. This creates a problem in that the data from different surveys are not directly comparable, and estimating $S_c$ over the whole profile from the available data is not straightforward (Kravchenko and Robertson, 2011).

Our approach to this is to use a Bayesian hierarchical model to estimate $S_c$ as a function of depth. There are strong theoretical grounds for expecting an exponential decline in soil carbon with depth. Firstly, the biomass of plant roots declines exponentially with depth, and this is a major source of carbon input via exudates and root death. Secondly, plant litter is deposited at the soil surface, and moves downwards in a quasi-stochastic manner (via soil macrofauna, disturbance events and leaching), which will produce an exponentially decreasing vertical distribution. The theory is borne out by empirical data: Jobbágy and Jackson (2000) found a logarithmic relationship between soil carbon stock and depth in a meta-analysis of more than 2700 soil profiles. Soil carbon stock is itself the product of two variables with sizeable sampling error (bulk density and carbon fraction, Equation 4) so would be expected to show a lognormal distribution. This is also borne out by the analysis of Jobbágy and Jackson (2000), who found that a logarithmic transformation of carbon stock gave the best linear model with depth, according to various diagnostic criteria.

Clearly, one could model this relationship using ordinary linear regression. However, a Bayesian hierarchical approach has several advantages. Firstly, it allows us to "borrow strength": by correctly representing the structure of the data, we can incorporate data from sub-populations of similar but distinct groups (e.g. data from different sites, locations, or surveys). In this way, it produces more accurate estimates, because it distinguishes between artefacts of the sampling process, rather than assumimng all data points are the same and lumping all unexplained variance into a single error term. As such, it can account for the fact that replicate soil cores from the same location at the same site are not independent samples, but will tend to be similar because of their co-location (and therefore have similar systematic differences from the global mean). Similarly, it can account for vagaries of different surveys, acknowledging that the errors and biases are likely to be similar in data from the same survey, because of systematic differences in protocols, laboratory procedures and instrument calibrations. For example the temperature and duration in the combustion furnace used can have a systematic effect on apparent carbon content measured by loss-on-ignition (Hoogsteen et al., 2015), but such vagaries are not explicit in the data typically available.

Secondly, the Bayesian approach allows us to propagate the uncertainty correctly, because we quantify the joint posterior distribution of all the parameters that we estimate (Gelman et al., 2013). Thirdly, and it allows us to bring in information from other sources as informative priors. For example, we can use the meta-analysis of Jobbágy and Jackson (2000) to set sensible values and ranges on the prior estimates for the parameters describing the relationship with depth. We can thus develop a statistical model which allows us to incorporate disparate data sets into coherent estimates of whole-profile carbon





stock, propagating the associated uncertainty appropriately. Critically, we can make use of the many observations collected at shallow depths, and use these to improve the estimates of whole-profile carbon stock, and the effects thereon of land use, whilst representing the uncertainty associated with the fact that these are *not* measurements of whole-profile carbon stock.

## 1.2 Validity of space-for-time substitution

We can use a number of approaches to investigate the validity of the assumption behind the space-for-time substitution.

1. We would expect any increases in $S_c$ which are caused by land-use change to be independent of the initial $S_c$. By contrast, we expect any pre-existing differences within a soil series to be in relative terms: a given soil series might have a high mean $S_c$, and the differences between land chosen for arable cropping and forestry will be proportional to this. We can therefore examine whether the ostensible effects of land use are consistent with models of absolute or relative change. The degree of relative change can be equated with pre-existing difference, and the confounding effect removed from the space-for-time substitution.

2. Along similar lines to the above, we can explore correlations between altitude and $S_c$, removing the effect of land use. Altitude affects climate and soil formation, and this strongly influences choices made over land use, such that different land uses lie on a gradient: as altitude increases, there is a general shift from arable crops to improved pasture, to rough grazing and forestry. If there are pre-existing differences in $S_c$ along this gradient irrespective of land use, then this can be estimated and the confounding effect accounted for.

3. In a Bayesian approach, we can use prior information in combination with observed data to improve our estimates. In the current context, we have information from previous paired-plot studies and modelling studies, which give us an expectation of the likely magnitude of land-use effects. These can be used to counter-balance artefacts which may exist in the space-for-time survey-based estimates of these effects.

## 1.3 Aims

Our aims here were to improve the estimates of the effects of land use on soil carbon stocks in the UK by:

– using more recent survey data to update the estimates of Bradley et al. (2005);
– developing a Bayesian hierarchical model whereby data from different surveys and sampling depths can be incorporated into estimates of whole-profile carbon stock, propagating the associated uncertainty appropriately; and
– investigating the validity of the assumption behind the space-for-time substitution.





## 2 Methods

### 2.1 The measurement of soil carbon stock and notation

Naming of the relevant quantities involved in measuring soil carbon is rather inconsistent, and it is useful to define some notation. Sampling is usually done by extracting a soil core with an auger, or less commonly, by digging a soil pit (see Nayak et al. (2019) for a recent review of field and laboratory methods). This provides a sample with known area $(A, \mathrm{m}^2)$ which is typically divided into one or more depth intervals. These intervals are defined by upper and lower boundaries $z_1$ and $z_2$, with length $d = z_2 - z_1$ in m. The soil from each depth interval is treated as a separate sample for laboratory analysis. The common laboratory process is to sieve the soil through a 2-mm sieve, removing stones, roots and any identifiable litter or living material, to leave the fine earth component (subscript $_{\mathrm{fe}}$). The total mass of fine earth $(M_{\mathrm{fe}}, \mathrm{kg})$ in each sample is oven-dried, weighed and expressed per unit volume to give the bulk density:

$$\rho_{\mathrm{fe}} = \frac{M_{\mathrm{fe}}}{dA} \quad [\mathrm{kg\,m}^{-3}]. \tag{1}$$

The organic carbon content of the fine earth is determined on a very small sub-sample, typically less than 10 g when using the loss-on-ignition (LOI) method, or <1 g when using the elemental analyser method, so the soil needs to be ground and well mixed to homogenise, and replicates taken. In the LOI method, a dried sub-sample is weighed, then heated to around 350-500°C for several hours (Ball, 1964, Hoogsteen et al. (2015)) so that the organic matter is combusted to $CO_2$ gas and water vapour. The sample is re-weighed, and the mass loss attributed to organic matter; around 55 % of this is carbon, depending on the chemical composition of the organic matter. In an elemental analyser, the combustion gas is trapped, and the exact amounts of $CO_2$ and $H_2O$ determined, so that the carbon loss is measured directly. Often, elemental analysis is used to verify or calibrate the carbon fraction determined by LOI. Depending on the analytical setup, the total (as opposed to only organic) carbon content may be measured, from which the inorganic fraction has to be estimated, and this may be significant in UK soils. The mass of carbon $(M_{\mathrm{c}})$ is expressed as a fraction of the mass of fine earth to give the carbon fraction (also referred to as carbon content or concentration):

$$f_{\mathrm{c}} = \frac{M_{\mathrm{c}}}{M_{\mathrm{fe}}} \quad [\mathrm{kg\,kg}^{-1}]. \tag{2}$$

The product of this and bulk density gives the density of carbon:

$$\rho_{\mathrm{c}} = \rho_{\mathrm{fe}} f_{\mathrm{c}} \quad [\mathrm{kg\,m}^{-3}]. \tag{3}$$

The quantity we are ultimately interested in is the mass of carbon per unit ground area i.e. the areal density, more commonly referred to as the carbon stock. This quantity always has some implicit value of depth associated with it. We may refer to the areal density of fine earth or carbon *within* a given depth interval $i$:





$$s_{\text{fe},i} = d_i \rho_{\text{fe},i} \quad [\text{kg m}^{-2}] \tag{4}$$

$$s_{\text{c},i} = d_i \rho_{\text{c},i} \quad [\text{kg m}^{-2}]. \tag{5}$$

and denote this with lower-case $s$. We can accumulate these with depth through the soil profile to give the cumulative areal
density down to a given depth $z$:

$$S_{\text{fe},z} = \sum_{i=1}^{n} s_{\text{fe},i} \quad [\text{kg m}^{-2}] \tag{6}$$

$$S_{\text{c},z} = \sum_{i=1}^{n} s_{\text{c},i} \quad [\text{kg m}^{-2}] \tag{7}$$

and denote this with upper-case $S$; $n$ is the number of intervals above depth $z$. As discussed below, there are statistical issues
in estimating the cumulative quantity in this way.

The areal density $S_{\text{c}}$ may be calculated for a specified depth $z$, or for a specified mass of fine earth $M_{\text{fe}}$ (i.e. spatial or
cumulative mass coordinates, *sensu* Gifford and Roderick (2003), and see Ellert and Bettany (1995)). The latter is important
for removing confounding effects of changes in bulk density (or sample compression) when only the upper layers of soil are
measured, on the assumption that the areal density of fine earth $S_{\text{fe}}$ does not change, even if the bulk density $\rho_{\text{fe}}$ does change
(Toriyama et al., 2011). The distinction is less critical if meaurements effectively encompass the bulk of the soil profile (down
to ~0.6 to 1 m): if a large majority of the soil containing organic material is sampled, then it matters less whether this is
expressed in spatial or cumulative mass coordinates. More critically, we want a method which allows us to combine data from
surveys which use different sampling depths, both shallow and deep. We describe this below.

### 2.2 Model development

Our approach was to build a parsimonious statistical model which explains the variability in the observations of soil carbon as
a function of land use and depth. Often the cumulative soil carbon stock $S_{\text{c},z}$ is used as the response variable, but this violates
the assumption of independence among samples. The value of $S_{\text{c},z}$ at each depth is contingent on all values at shallower depths
being correct, so this underestimates the uncertainty. By using the variables that are actually measured, we can more correctly
represent the structure of the data, and thereby more accurately quantify the associated uncertainty.

For each soil core $i$ from land use $u$ occurring at location $j$ within site $k$, we predict the carbon density $\rho_c$ as a linear function
of the sampling depth $z$, according to:



$$\log(\rho_{c,z,u})_i \sim N\left(\mu, \sigma^2\right)$$

$$\mu = \beta_{0,u} + \beta_{1,u}z + b_{0j[i],k[i]} + b_{1j[i],k[i]}z$$

$$\begin{pmatrix} b_{0j} \\ b_{1j} \end{pmatrix} \sim N\left( \begin{pmatrix} \mu_{b_{0j}} \\ \mu_{b_{1j}} \end{pmatrix}, \begin{pmatrix} \sigma^2_{b_{0j}} & \rho_{b_{0j}b_{1j}} \\ \rho_{b_{1j}b_{0j}} & \sigma^2_{b_{1j}} \end{pmatrix} \right), \text{ for loc\_id:site\_id } j = 1,\ldots,\text{J}$$ (8)

$$\begin{pmatrix} b_{0k} \\ b_{1k} \end{pmatrix} \sim N\left( \begin{pmatrix} \mu_{b_{0k}} \\ \mu_{b_{1k}} \end{pmatrix}, \begin{pmatrix} \sigma^2_{b_{0k}} & \rho_{b_{0k}b_{1k}} \\ \rho_{b_{1k}b_{0k}} & \sigma^2_{b_{1k}} \end{pmatrix} \right), \text{ for site\_id } k = 1,\ldots,\text{K}$$

The $\beta_0$ and $\beta_1$ parameters represent the global intercept and slope parameters specific to each land use ("fixed effects" in the mixed modelling jargon). The $b_0$ and $b_1$ parameters represent group-specific parameters describing the variability among sites, and at locations within sites ("random intercepts and slopes" in the mixed modelling jargon). These local deviations in intercepts and slopes account for the within-location/site correlation of residuals, and are assumed to be independently drawn from normal distributions with the means and covariance matrices shown above. Analagous terms can be added to account for systematic differences between surveys (i.e. data sources) but are not shown here for brevity. The only difference is that these are crossed effects, applying to all sites within a survey, rather than nested effects.

The parameters of hierarchical models are commonly estimated by maximum likelihood estimation (MLE). Here, we used MLE for exploratory analysis, but used a Bayesian method for the final model, for the reasons given earlier. Bayesian methods generally uses Markov chain Monte Carlo (MCMC) sampling, an iterative algorithm for calculating numerical approximations of multi-dimensional integrals. Many MCMC algorithms are available, and the mechanics of performing Bayesian statistical analysis are described in several textbooks (e.g. Gelman et al., 2013). Here we use the Hamiltonian sampling algorithm, which provides a computationally efficient means of estimating the posterior distribution (Betancourt, 2017; Stan Development Team, 2016) via the R package `brms` (Bürkner, 2021).

Equation 8 predicts carbon density, whereas the quantity we require is the carbon stock $S_{c,z}$ over a 1-m depth (as used by Bradley et al. (2005)). The prediction from the linear model at the mid-point $z/2$ yields $E[\log(\rho_c)]$, equivalent to the geometric mean or median $\rho_c$ over the depth $z$. However, the quantity equivalent to $S_{c,z}$ is $E[\rho_c]$ or the arithmetic mean $\bar{\rho}_c$. The inequality between these is well-known, and the "transformation bias" can be approximated in a number of ways (e.g. Miller 1984). Here, we use a simple but reliable method, whereby we numerically calculate $E[\rho_c]$ from a set of predictions from Equation 8 at 1-mm intervals over a 1-m depth. We then calculate the depth at which the arithmetic mean $\bar{\rho}_c$ occurs as

$$z_m = \frac{\log(\bar{\rho}_c) - \beta_{0,u}}{\beta_{1,u}} \quad [\text{m}]$$ (9)

where $\beta_{0,u}$ and $\beta_{1,u}$ are the estimated intercept and slope for each land use in Equation 8. Although these parameters vary by land use, so $z_m$ is not completely constant, the variation is only a few cm, and we can use the mean value of $z_m = 0.37$ m





for all with little loss in accuracy. Having established the depth at which the arithmetic mean soil carbon occurs, we can then use predictions from Equation 8 at $z = z_m$ where $E[\rho_c]$ is equal to $S_{c,z}$.

## 2.3 Data sources

### 2.3.1 Bradley et al (2005)

Bradley et al. (2005) collated soil core data from separate surveys covering England and Wales, Scotland, and Northern Ireland. Common depth layers were defined across all survey results (0–30 cm and 30–100 cm). Litter horizons were included but unconsolidated subsoil horizons and bedrock were excluded. The soils were classified according to four land-use types: crops and cultivated land (mainly arable); improved, managed grassland; grassland that received no management and semi-natural vegetation; and woodland. For brevity, we refer to these as crops, improved grass, rough grazing and woods, respectively.

These classes were chosen so that there would be a reasonable chance of having measured data that could be used for each soil series/land use combination, and the classes are broad enough that they can be applied across all the other surveys described below.

### 2.3.2 Countryside Survey

Countryside Survey is a unique study of the natural resources of the UK's countryside, carried out at approximately decadal

intervals since 1978 (Emmett, 2010, Robinson et al. (2020)). The survey uses a sampling approach which samples 1-kilometre squares randomly located within different land classes in GB. The original 1978 survey consisted of 256 1-km squares and collected five soil samples per square, taken from random co-ordinates in five segments of the square. There has been an increase in the number of sample squares over time, and in 2007, 591 1-km squares were sampled with a total of 2614 samples returned for analysis. The survey became a rolling programme in 2018 (Robinson et al., 2020). Within each 1-km square, a set

of soil samples were taken from five pre-determined random locations. A sample was taken by inserting a cylindrical plastic tube, 15 cm long, into the soil. Bulk density was measured on the extracted soil material after drying. The carbon fraction was measured using loss-on-ignition on a 10 g air dried sub-sample taken after sieving to 2 mm. The sub-sample was dried at 105°C for 16 hours to remove moisture, weighed, then combusted at 375°C for 16 hours.

### 2.3.3 Glastir Monitoring and Evaluation Programme (GMEP)

A total of 300 x 1-km squares in Wales were sampled between 2013 and 2016 as described by Robinson et al. (2019). The same protocol was used as in the Countryside Survey, described above.

### 2.3.4 Reading Agricultural Consultants surveys (RAC)

A large soil sampling campaign was conducted along the route of the High-Speed 2 (HS2) rail project (Heming, 2021), as well as at other locations in England, providing almost 2000 soil cores in total. Soil sampling procedures are described in Heming

et al. (2021), and report carbon in 0-25 cm and 25-50 cm layers. Soil carbon analysis used the elemental analyser technique.





### 2.3.5 Ward et al. (2016)

Ward et al. (2016) conducted a survey of 180 permanent grasslands sites located throughout England. Sampling sites were in 60 different geographical locations, from 12 broad regions of England. At each of the 60 locations, three different fields were selected to give a gradient of management intensity of extensive, intermediate and intensive management. Soil cores in 3.5 cm in diameter were taken from three random areas in each field, to 1-m depth using an Eldeman auger, and divided into five depth increments: 0–7.5 cm, 7.5-20 cm, 20-40 cm, 40-60 cm and 60-100 cm. The carbon fraction was measured using an elemental analyser, along with bulk density.

### 2.3.6 ELUM

27 sites were sampled for the Ecosystem Land Use Modelling (ELUM) project (Keith et al., 2011), covering bioenergy crops, arable crops, grasslands and woodlands. Soil sampling details were the same as for Ward et al. (2016), but the depth layers were 0-30 cm, 30-50 cm and 50-100 cm.

### 2.3.7 Easter Bush

Repeated intensive soil sampling has taken place at the Easter Bush site in central Scotland in 2004, 2012 and 2015 (Schrumpf et al., 2011, Jones et al. (2016)). Although this covers only a single site, we include the data here for the detailed characterisation of the variation in soil carbon with depth. At each sampling time, 100 cores were taken on a regular grid with 15-m spacing, sampling the whole profile depth (to around 60 cm). A corer with an inner diameter of 8 cm was used for extracting soil samples and bulk density measurements. Both loss-on-ignition and elemental analyer techiques were used.

### 2.3.8 Meta-analyses

Various published studies have collated data from the literature to perform meta-analyses of land-use effects on soil carbon. We compare results with several of these studies (Post and Kwon, 2000, Guo and Gifford (2002), Berthrong et al. (2009), Poeplau et al. (2011)).

## 3 Results

Figure 1 shows the spatial distribution of some of the soil survey data in relation to topography and previous mapped estimates of soil carbon stock. The data are more focussed on the southern half of the UK, where soil carbon stocks are lower than in the north. However, this is also the region where population pressure is more intense, pressures driving land-use change are stronger, so these may also be more pertinent to the analysis. Spatial data are not publicly-distributable for all the data used here because of non-disclosure agreements and availablility of raw data, which affects the Countryside Survey and Bradley et al. (2005) data. The spatial coverage in the Countryside Survey and Bradley et al. (2005) data are much more widely distributed, and numerically these dominate the data set.



Examples of the relationship between carbon density $\rho_c$ and depth in four soil cores, taken from different sites and surveys, are shown in Figure 2. With a logarithmic $y$ axis, a reasonably linear relationship is seen. Variants using logarithmic transformations of $x$, $y$ and both $x$ and $y$ axes, were explored (Figure S2 & S3). The best model in terms of variance explained and analysis of quantile plots was the one shown here. In organic soils, reasonable relationships were still seen, but $\rho_c$ was much more constant with depth, so the variance explained by a linear trend was less. In soils with complicated profiles, with organic

horizons within mineral soils, the fit was much poorer, but these soils were relatively infrequent.

The entire data set is plotted on these same axes in Figure 3, split by the main land uses. Clearly, there is considerable variation within the main land uses (particularly noting that the $y$ axis is logarithmic). Much of this accounted for by the group-specific terms in the hierarchical model, such that the model accounts for almost 90% of the variance. Around half of this is attributable to the effects of land use and depth, the remainder attributable to the site and location group-specific terms.

The interpretation of lhe latter is that there is consistent site-to-site and location-to-location variation in the data, which we can identify, and separate from the estimate of the population-level effect of land use. However, the group-specific terms do not represent "explained variance", in the sense that they are not useful for prediction outwith the sample. Ordinary least-squares regression lines are shown for each survey within these groups to identify the broad differences between surveys. The surveys give broadly similar results, but some systematic differences seem to be apparent. The results from Ward et al. (2016) sit higher

than the others for grassland; the results from the ELUM survey sit lower. The slope with depth was generally greater in the data of Bradley et al. (2005) than the RAC survey data, but the mean values were very close. No surveys measured at the same sites, so separating these effects in the model is difficult. With a site-specific effect already accounted for in the model, an additional survey-specific effect was not warranted, based on standard model selection criteria.

To estimate the mean soil carbon stock for each land use, we use predictions from the model over a depth of 1 m (at $z_m$

where $E[\rho_c] = S_{c,z}$), excluding the site- and location-specific terms (which average out to zero at population level). These are shown in Figure 4 with associated 95% confidence intervals and prediction intervals. Soil carbon stocks are highest in woods, followed by rough grazing, and improved grasslands, with arable crops having the lowest values. The differences among land uses are larger than the 95% confidence intervals in all cases, so we can consider that these are real, discernible effects. The prediction intervals, which represent the variability in predictions of the effect at a site outwith the sample, are much larger

than the effect size.

The estimated effects of land use are expressed in Figure 5 as the difference from the mean soil carbon stock. The effects are estimated from each survey individually, as well as using all the data. In almost all cases, the effect is of similar magnitude and sign. The sign of the effect appears more variable in improved grasslands, but this lies near the mean, and the degree of variablilty between surveys is similar to elsewhere, so values may be above or below the mean. The 95% confidence intervals

generally do not include zero, so we can be reasonably sure of the overall effects. The predictions intervals, however, span positive and negative values, and are much larger than the effect size.

Also shown are results from three other sources: predictions from the IPCC Tier 1 default method for comparable land-use types; the meta-analysis of Guo and Gifford (2002); and the mean effects from the synthesis of (mostly) paired-plots studies by Poeplau et al. (2011). These all show similar effect sizes to each other, and of consistently the same direction as the UK data.



The magnitude of the effect size in arable crops is also very similar to the UK data; that in improved grasslands is close, but
consistently positive, whereas the UK data is on average lower than the mean (but as noted above, both lie close to the mean).
The effect size for woods is considerably larger in the UK data than in the meta-analyses. The comparison for rough grazing is
more restricted, as there is no comparable land use in the meta-analyses, but the IPCC default values are rather lower than the
UK data.

Table 1 shows the results of this study compared to the results reported by Bradley et al. (2005), in terms of the effect of land
use on soil carbon stock. Because mean values differ between studies, we use the value for arable crops as the reference level,
the land use with lowest carbon stock. This does not affect the magnitude of the effects, but ensures all effects appear positive.
Our values are 1.3, 1.7 & 3.4 times smaller than those of Bradley et al. (2005).

In Figure 6, we examine whether the ostensible effects of land use are consistent with models of absolute or relative change.
This shows the apparent size of the land-use effect with increasing soil carbon stock in the data of Bradley et al. (2005). This is
expressed as the difference from the value for arable crops on the same soil series, so that the overall effects are positive in all
three cases. If differences caused by land-use change were independent of the initial $S_c$, we would expect the black horizontal
line, indicating no change in the mean effect size. If there were pre-existing differences within a soil series between land chosen
for arable cropping and (say) forestry, these are expected to be proportional to the soil carbon stock (because the variance in
$S_c$ is typically proportional to $S_c$). This would cause the apparent effect of land use to increase with $S_c$, and this is what we
observe in Figure 6. The slope is greatest in woods and least in grass, so the magnitude of any error will vary among land uses.





**Figure 1.** Spatial distribution of a subset of the soil survey data in relation to altitude (left-hand panel) and previously mapped soil carbon stock from Bradley (2005) (right-hand panel). Spatial data cannot be made publicly available for the Countryside Survey and Bradley (2005) raw data, where the spatial coverage is more evenly distributed across the country.





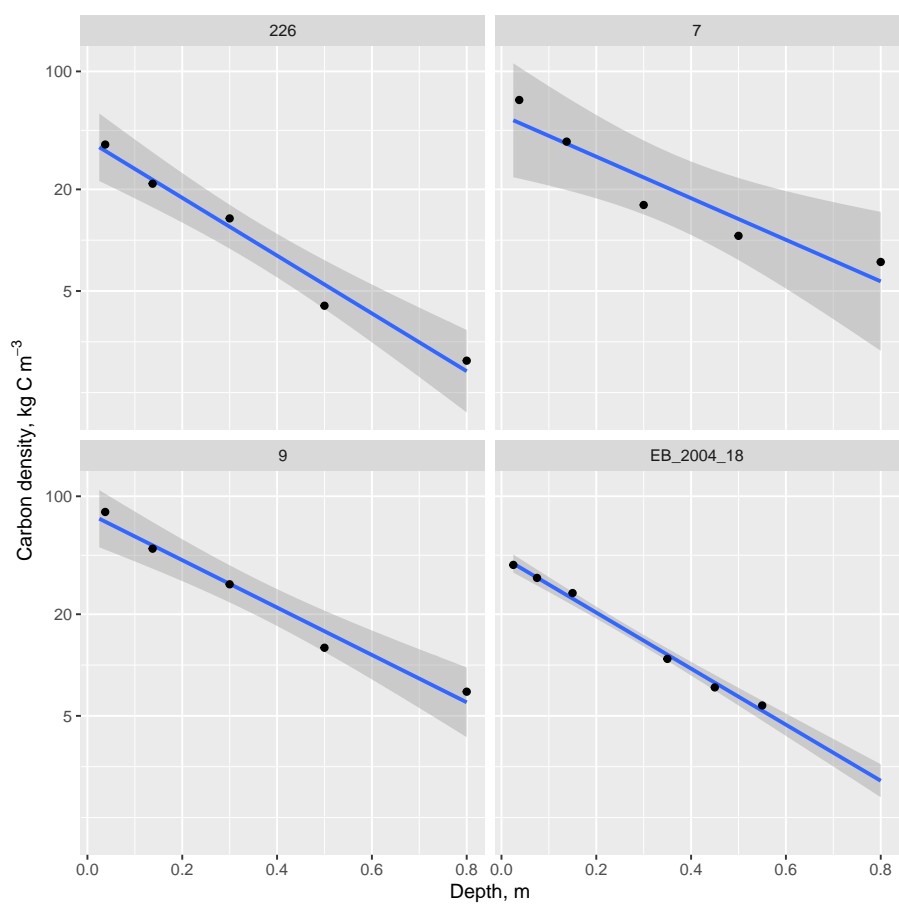

**Figure 2.** Four examples of the relationship between carbon density $\rho_c$ and depth in soil cores from various sites. Note the logarithmic $y$ axis scaling. Of the various simple models using different transformations, this gave the best linear relationship. The soil cores come from the surveys of Ward et al (2016) (samples 226 and 7), ELUM (sample 9), and Easter Bush (sample EB-2004-18).

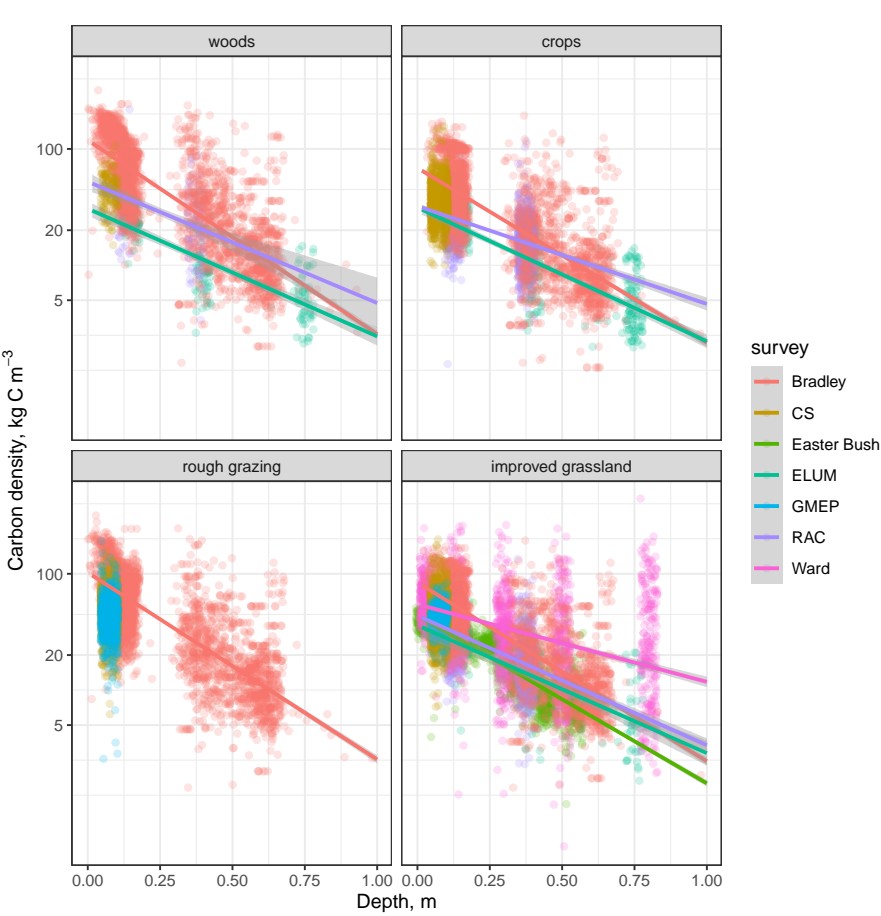

**Figure 3.** Relationship between carbon density $\rho_c$ and depth by land-use type, showing all data. Note the logarithmic $y$ axis. Ordinary least-squares regression lines are shown for each survey.





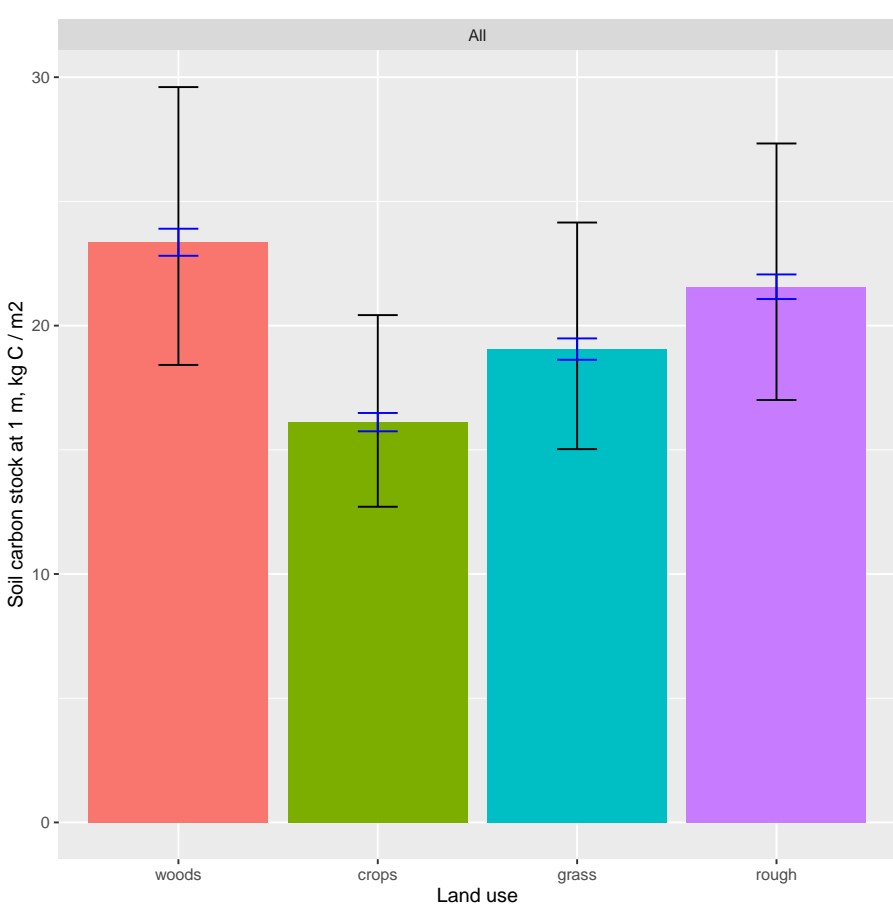

**Figure 4.** Predicted mean soil carbon stocks to a depth of 1 m for each land use, as estimated by the hierarcical model using all data. Associated confidence intervals are shown as blue error bars, prediction intervals are shown in black.



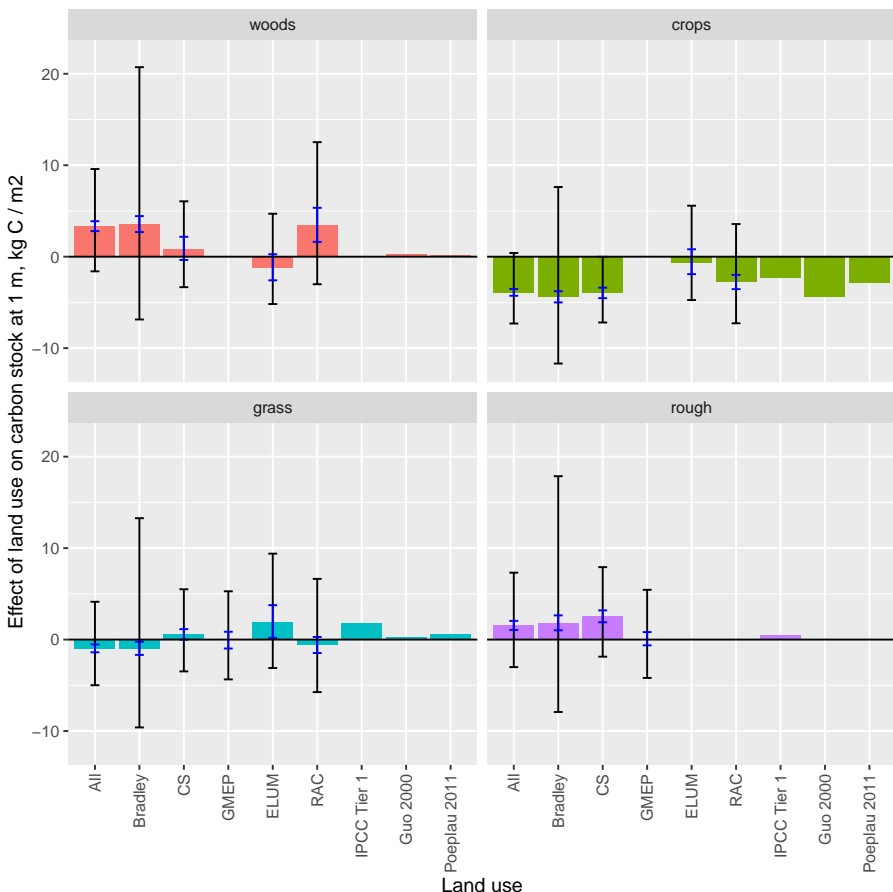

**Figure 5.** Predicted effect of land use on soil carbon stocks to a depth of 1 m, as estimated by the hierarcical model, expressed relative to the mean soil carbon stock. Associated confidence intervals are shown as blue error bars, prediction intervals are shown in black. Values are shown for each survey individually, as well as the full data set. In addition, we plot comparable predictions from the IPCC Tier 1 model, and the meta-analyses of Guo and Gifford (2002) and Poeplau et al (2011). For the latter in the case of rough grazing, no comparable values were available, so these are missing rather than zero.

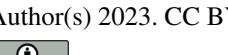


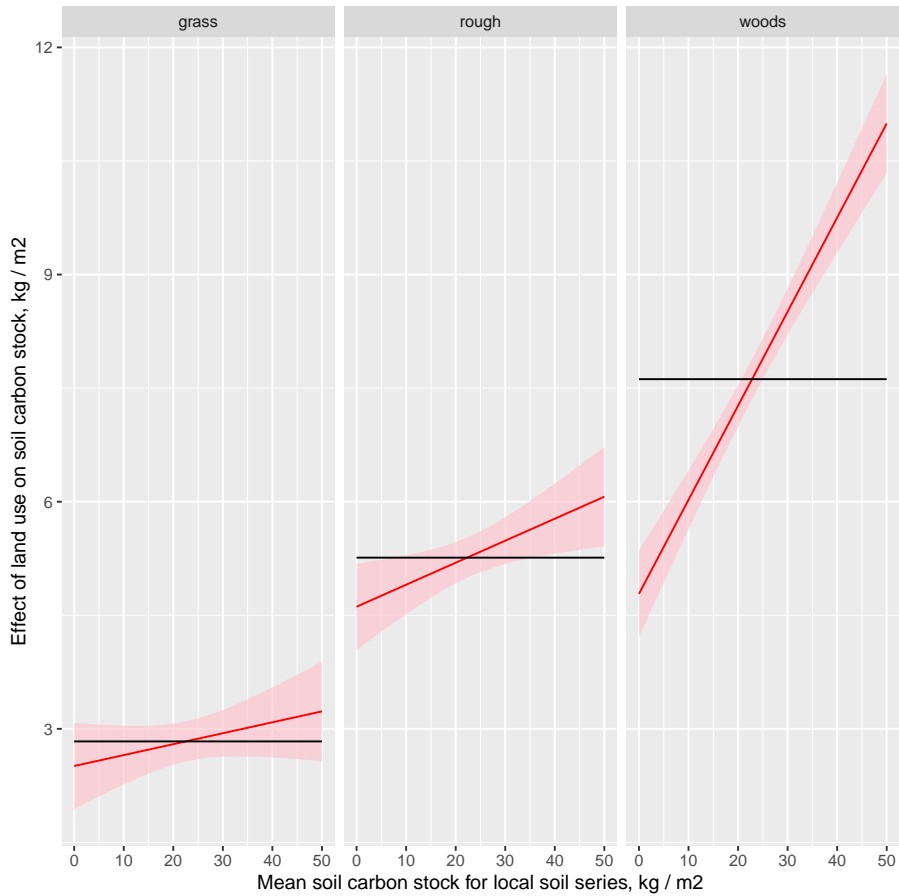

**Figure 6.** The apparent relationship between the effect of land use and increasing soil carbon stock in the data of Bradley et al (2005). The effect is expressed as the difference from the value for arable crops on the same soil series, so that the overall effects are positive in all three cases. The black horizontal line indicates an expected constant mean effect size in the absence of any effect. Red lines show ordinary least-squares regression lines fitted to the data. The individual data points are not shown because they are highly variable and so require a much larger *y* axis scale for display.



**Table 1.** Estimates of the effect of land use on soil carbon stock from this study, with 95% confidence intervals, compared with the estimates reported by Bradley et al (2005). Effects are expressed relative to arable crops, the land use with lowest carbon stock, so all effects appear positive. Units are kg C/m2.

| Land use | Bradley et al (2005) | This study | 95% CI, this study |
|----------|---------------------:|-----------:|--------------------|
| woods    | 13                   | 7.24       | 6.7 to 7.79        |
| grass    | 4                    | 2.94       | 2.51 to 3.37       |
| rough    | 20                   | 5.45       | 4.96 to 5.95       |



## 4   Discussion

The results here have improved the estimates of the effects of land use on soil carbon stocks in the UK by increasing the amount of data they are based upon, and providing a more sophisticated analysis which allows the incorporation of soil samples
measured at any depth. The Bayesian hierarchical approach allows us to compensate for biases in the choice of sites and protocols used in different surveys, and to propagate the associated uncertainty appropriately.

Because of the large sample size, the data of Bradley et al. (2005) still carry a lot of weight in the overall estimates. However, the new estimates are substantially smaller than the reported estimates of Bradley et al. (2005). We identify a number of factors behind this. The main factor is simply because we use a logarithmic transformation of the carbon density data. This is done for
several reasons: it gives an approximately linear relationship with depth, it stabilises the variance so that it is relatively constant with increasing $\rho_c$, and makes the model residuals close to normal. The effect is to give less weight to extreme values. Implicit in using a logarithmic transformation is an assumption that the variance is constant in relative, not absolute terms. Whilst the log transformation is a fundamentaly better model in this case (because the assumption of relative variance is borne out by the data), it is not a perfect model, and other options are possible: different transformations, Box-Cox transformation, a generalised
linear model with Gamma-distributed residuals. All of these would give somewhat different answers, with different weights given to extreme values, and in reality there remains epistemic uncertainty as to the best way to summarise the data in the form of a model. A more comprehensive analysis could apply several of these approaches and use a model averaging approach to indicate the posterior uncertainty that is introduced by this. Clearly this can be considerable, comparing the results with the previous analysis on the untransformed data.

A second factor is in the analytical method used by Bradley et al. (2005), which implicitly weighted the effect from each soil series by its area. This would give more weight to the soil series with the largest expanse, although these may have little or no land-use change (e.g. in large areas of the uplands). If this effectively gives more weight towards areas with high soil carbon stocks, and the effect of land-use change is over-estimated in these areas for the reasons discussed earlier, then this could introduce a bias. Our approach does not reweight the data based on estimated areal coverage, and attempts to estimate
the overall effect of land use, having accounted for the vagaries of different sites and soil series. This is a slightly different approach, but should give the best estimate of the effect at a new location outwith the sample.

The results suggest that assumption underlying the space-for-time substitution is not completely valid, although they are not unequivocal. The apparent increase in the effect of land use with increasing soil carbon suggests that there are pre-existing differences which confound the comparison, and will lead us to over-estimate the effect. This over-estimation varies across
land uses, greatest in woods and rough grazing (~30% and 15%, respectively). This tallies with the comparison with meta-analyses of paired-plot studies, which show smaller effect size. However, the difference is much less in improved grasslands and arable crops, so the effect of pre-existing differences may be much less in these cases. Exploring the trend in soil carbon with altitude, which might lead to an expectation of pre-existing differences in land selected for different uses, showed no strong relationship (although this did not include the Bradley et al. (2005) or the Countryside Survey data). Long-term single-
site experiments where changes in land use have been recorded over several decades are still very valuable as a reference point



for other analyses. (eg. the Rothamstead Park Grass experiment etc.). The Countryside Survey data are important as one of the few long-term surveys with repeated measurements at a spread of locations wide enough to be used to infer national-scale trends. Although these did not include bulk density in the early surveys, so we cannot make strong inferences about $S_c$, we can be reasonably confident that there is no substantial change in carbon fraction (Figure S6).

The results show some systematic differences between surveys in estimates of carbon stocks, and this is to be expected for various reasons. Bradley et al. (2005) included all litter horizons in their definition of soil, whereas CS and GMEP included only the F (folic, partially decomposed) and H (humic, decomposed) layers, and excluded the readily recognisable litter. Differences in the attribution of litter material to the soil have been noted before as a key difference between soil survey schemes, and an important factor for consistency in experimental comparisons, particularly in woods and forest environemnts (Poeplau et al.,

2011). This may explain why the data of Bradley et al. (2005) show higher carbon densities in woodlands compared to CS, with the greatest difference in woodlands. The effect is less pronounced in other land uses where there is less litter. This may explain some of the difference shown in the comparison of the effect of woodland land use among surveys (5), and in Table 1, although the effect of log transformation is the dominant difference here. There will also be differences among surveys caused by differences in laboratory analytical procedures. Various authors have shown effects of sample mass, combustion

temperature, and duration on the mass loss observed in the loss-on-ignition method, particularly highlighting the extent of structural water loss and its dependency on clay content (e.g. Hoogsteen et al., 2015). Heming (2021) reject the method on the basis of the uncertainty introduced by the latter. Similarly, the elemental analyser approach is prone to the uncertainties associated with separating organic and inorganic carbon (Wang et al., 2012), and this will introduce a different set of biases. Consistency is key in long-running survey schemes, so that true changes over time can be separated from changes related to

different analytical methods. The hierarchical statistical method used attempts to remove such effects, partitioning them into the group-level effects, and not part of the overall effect of land use which we are trying to discern.

Soils vary widely in depth, and the modelling approach used here makes the comparison across a standardised depth of 1 m, based on the guidelines for UNFCCC reporting. Many soils in the UK are shallower than this: around half the soils in Wales are in classifications with a lithoskeletal substrate, and defines the soil to be less than 80 cm thick. We note that the results should

be interpreted as predictions of the effect of land use on a typical but hypothetical soil of 1-m depth, taking into account the way in which carbon density declines with depth. A more complex model would be required to make accurate spatial predictions.

With the large data set analysed here, the uncertainty in the differences among land uses was small enough to identify consistent mean effects. However, a striking feature of the results is the extent of the variability in these effects, represented by the large prediction intervals, and this was similar across all surveys. Mapping soil carbon using geostatistical or machine

learning methods has become widespread and increasingly accurate because of the strong spatial relationships in the data (Hengl et al., 2017). However, this may lead to a false sense of understanding of causal relationships. The data here highlight the degree of variability, and the effects of land use, whilst discernible in very large samples, are not as clear as is generally pre-supposed (Baker et al., 2007, Kravchenko and Robertson (2011)). This has important consequences for attempts to verify schemes which aim to sequester carbon in the soil by altering land use.





With pressure to find measures which will help meet targets for net-zero emissions, there is now considerable interest in options for sequestering carbon in agricultural soil (Soussana et al., 2019, Alliance (2022)). To be credible, these options will need to be verifiable, and governed to ensure permanence and prevent leakage or reversals (Smith et al., 2020, Black et al. (2022)). If soil carbon credits are to be used to pay farmers for changes in land use or management, the effects need to be demonstrable over relatively short time scales and at a practicable cost. The results we find here have implications for the prospects for this. Although we can demonstrate an overall mean effect greater than the uncertainty bounds with a large sample size (more than 25,000 core sections at several thousand locations), we find that the variability in these effects is much larger. The prediction intervals were much larger than the mean effect size, and spanned both positive and negative effect sizes. This result was consistent across all the surveys we analysed.

In any given instance where we wish to verify the change in soil carbon, our expectation is that the observed change in soil carbon will lie in a very wide range, encompassing both possible gain and loss. Indeed, there will often be instances where the observed effect is opposite to the expected mean effect. The effect of a given intervention is therefore very hard to verify, and this raises questions over how this might be handled in such schemes as a formal part of efforts to mitigate climate change.

The effect of the changes to the estimates of equilibrium soil carbon on the carbon fluxes arising from land-use change is not obvious. The changes will reduce the gross fluxes from each land-use change, but how this affects the net effect depends on the balance of changes and the transition matrix of land-use change each year (Levy et al., 2018), so the net effect is not linearly predictable, and requires further analysis.

## 5   Conclusions

We have produced new estimates of the effects of land use on soil carbon stocks in the UK. These are smaller than the previous best estimates of Bradley et al. (2005), partly because of new data, but mainly because the effect is more reliably characterised using a logarithmic transformation of the data. We characterised the uncertainty and variability in the effect. With the very large data set analysed here, the uncertainty in the differences among land uses was small enough to identify consistent mean effects. However, the variability in these effects was large, and this was similar across all surveys. This has important consequences for attempts to verify schemes which aim to sequester carbon in the soil by altering land use. Whilst we can estimate the expected overall effect at national scale, the effect in any given instance is expected to lie in a very wide range. The effect of a given intervention is therefore very hard to verify, and may be difficult to include as a formal part of efforts to mitigate climate change. Examining whether the "space-for-time" substitution is valid, the results were not unequivocal, but we estimated that the effects are likely to be over-estimated by 5-33%, depending upon land use.

*Code and data availability.*   Several of the data sets are available via the UK Environmental Information Data Centre (EIDC) at https://eidc.ac.uk/.



*Author contributions.* PL performed the modelling analysis and wrote the manuscript. All other authors provided data and contributed to the manuscript.

*Competing interests.* The authors declare no competing interests.

*Acknowledgements.* This work was supported by the Natural Environment Research Council through award number NE/R016429/1 as part of the UK-SCAPE programme delivering National Capability and the AgZero+ project. The Glastir Monitoring and Evaluation Program (GMEP) was funded by the Welsh Government as part of the Environment & Rural Affairs Monitoring and Modelling Programme. David Robinson was funded in part by funding from the European Union Horizon Europe research and innovation programme under grant agreement No.-101086179, funding from the UK Research and Innovation (UKRI) under the UK government's Horizon Europe funding guarantee [grant number 10053484], and the Research Council of Norway, Climasol, Project number: 325253.



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
