# Peer review of "The Effects of Land Use on Soil Carbon Stocks in the UK"

_EGUsphere, 2023_

## Author Response (AR1)

**Reply to referees' comments**

Peter Levy

2024-07-12

We thank the referees for their time taken. Their comments are shown in *italics*; our response is beneath in normal font.

**1 Referee 1**

*Narrative on space-for-time: I think that the space-for-time assumptions affecting land use change impacts can come earlier in the abstract and introduction. After the second sentence of the abstract, an immediate question that came to my mind was how time affects land use change impacts on soil C. I wondered this again by line 27 of the introduction. Since this paper attempts to deal explicitly with space-for-time assumptions, I suggest highlighting that aspect early on.*

- we introduce the space-for-time assumption on line 13 in the first paragraph of the Introduction. It cannot really come any earlier.

*Intro: condense 3 sections into one to make the writing more concise. Sections 1.2 and 1.3 repeat the prose of section 1.*

- no, we feel a limited amount of repetition is useful.

*Methods: start with brief description of each dataset used in the manuscript.*

- no, we need to introduce the general method and notation before we can go into the specifics of each data set. Otherwise, it would not be clear why it is important if a data set is missing bulk density measurements, or is limited to topsoil.

*Methods are a bit difficult to follow. Perhaps organize according to the three main approaches to improve estimates outlined in lines 56-59?*

- we already have sections on two of these approaches; we have added the third (space-for-time) as suggested.

*Model – how is model fit determined?*

- also requested by other referees - we have added details at the end of the 2.2 Model Development section. We have added model performance metrics as a Table (new Table 1) and some further description of the model selection process in the Results.

*Is the depth distribution of Sc modeled as part of the Bayesian framework?*

- yes, this is explicit in Eqn 8 and throughout the text.

*If yes, is there a need to control feedbacks in the model so that information only flows one way (eg: estimates of main LU effects do not feedback to the depth distribution? See Ogle et al 2013 and Ogle and Pendall 2015).*

- no, the model structure is simple enough that this is not an issue: we are fitting intercept and slope terms for each land use; these *are* the main effects.

*The results of this paper show smaller soil C stock estimates compared to Bradley. Would it be possible to run the analysis only on the Bradley data to get some sense for the extent to which the change is due to modeling choices and the extent to which it's due to a larger and more extensive dataset?*

- that comparison is already shown in Fig 5. We already cover this in the first page of the Discussion; the difference is largely due to modelling choices rather than the additional data.

Results:

*Is the model fit reported anywhere? I may have missed this.*

- no, and the same point was made by referee 2. We have added this as a Table (new Table 1) and some further description of the model selection process in the Results.

*Develop the narrative by giving context first and then describing results. Eg, line 248: 'Figure 1 shows some of the soil survey data...' what is some? What's the question this figure addresses or what patterns should the reader pay attention to?*

- This is a good point - we added a map to show the wide coverage, and out of habit. However, given that we cannot diclose the locations of several of the large data sets, Fig 1 rather fails in this respect, as pointed out by the other referees, and is not really needed. We have removed it in the revision.

*Lead with a statement that directs the reader to the figure (Figure X). Eg, line 276 instead of 'these are shown in Figure 4', end the previous sentence with (Figure 4).*

- noted. We have added the suggested variety in the revision.

*Given figure 3 with all the data by depth, is figure 2 for specific cores, necessary? What does figure 2 add that cannot be seen in figure 3?*

- Fig 2 shows that the linear relationship is a close fit in individual cores. This point is lost when all cores are plotted together in Fig 3.

*In the figures showing prediction lines, are these lines drawn based on the bayesian model parameters?*

- yes, this is now explicit in the figure captions.

*Discussion: discuss the importance of showing confidence intervals and prediction intervals? Depending whether confidence or prediction intervals are interpreted, the conclusions would be substantially different, particularly for results shown in figures 4 and 5.*

- We discuss this in lines 367-387 so is already a main point in the discussion.

*Figure 1: explain abbreviations in caption. Label axes as latitude and longitude? The left panel has two color scales that are both spatial and I am unsure how to read the graph.*

- now redundant since we have removed Fig 1 as unnecessary.

*Figure 2: Do the points in this figure represent an average or individual points? If average, is there an error or uncertainty associated? If individual samples, why were these chosen? What are the titles of each facet (226, 7, 9, ED_2004_18)? include log in the y-axis name? Consider a flipped-axis in which the x-axis is shown vertically and the y-axis is shown horizontally? I think it's a more intuitive way to look at a variable like depth.*

- these are individual points, chosen to illustrate the typical range of this relationship; we have changed the caption wording slightly to make this clearer.

*Figure 3: include log in the y-axis name? Consider a flipped-axis in which the x-axis is shown vertically and the y-axis is shown horizontally?*

- The values shown are in original units; it is the axis scaling that is transformed here, as stated in the caption. Personally, I dislike flipped axes where y is the dependent - we have conventions for a reason.

*Figure 5: 'For the latter' includes which studies? Only Guoo and Gifford and Poepllau or also ELUM and RAC? To clarify, perhaps add an NA or ND to the figure itself where no data are available.*

*Figure 3-6: keep LU categories in the same order in all graphs.*

- noted to be chaged in the revision.

*Minor Line 10: What does 'This' refer to? It reads as if 'this' is the variability in mean effects, but I think it's meant to refer to the results on how land use ranks in soil C storage?*

- it refers to the variability, and we have changed this to be explicit "However, the variability in these effects was large, and this variability was similar across all surveys".

*Line 30: Why say >400 instead of listing an exact number?*

- because it is the magnitude that is important, not the exact number. The exact number depends on how you count them (all cases, series with some missing values etc.).

*Line 30: does this need a citation? The following paragraph implies that the samples from >400 soil series come from a particular dataset if other data has been collected since 2005.*

- This has been cited at the end of the previous sentence. We are still in the same paragraph, so still talking about the same thing, so we would not cite it again.

*Line 33: 'These data' – I got a bit lost as to which data is being refered to. The surface soil data collected since 2005?*

- yes. Changed to be explicit: "these more recent data".

*At this point in reading I am not sure if this paragraph leading the reader to the fact that the current study is going to use data since 2005 to update estimates or that data since 2005 can't be used ?*

- the point is they have not been used (until now) because no one had a method to do so.

*Line 34: 'there are some issues' is vague. Perhaps reword sentence to be more specific? Eg: 'Assumptions inherent in previous estimates (Bradley et al 2005) may have important limitations for interpretation of soil C stock changes.'.*

- we say what these issues are in the next sentence.

*Line 37: 'some of the details. . . with respect to land use are unclear' – is it possible to be specific about what exactly is unclear?*

- we have expanded this in the revision. There are no equations in their paper and no surviving source code of their data analysis, so there is a general lack of clarity in how one interprets the paper into a reproducible mathematical/code form.

*Line 38: is there a citation for choices/assumptions giving different results? Or are the authors stating that it's important to be more clear in the assumptions? Or even to run multiple models that vary assumptions in a type of sensitivity analysis?*

- all of these things. I do not think the general point needs a citation, but we show the importance here in Table 2, which shows the effect of choosing log-transformation or not.

*Line 39: keep with previous paragraph as this is about limitations in prior studies? Does 'the general approach' reefer to what Bradley did? Or more broadly, to soil stock/LUC analyses?*

- this refers more broadly to any soil stock/LUC analysis which uses the space-for-time substitution.

*Line 53: Is the Countryside Survey data the data referred to in line 30? Perhaps briefly describe the Countryside dataset? Many readers won't be familiar with it.*

- not sure why the comment for line 53 refers to line 30. Assuming this is intended:
- no, line 30 is a continuation from the previous sentence, expanding upon the data of "Milne and Brown, 1997, Bradley et al. (2005)" (line 28).

*Line 57: point i), the way that lines 31-34 are written, it seemed as if the more recent data are difficult to incorporate for analysis. Perhaps re-phrase line 31-34 to better lead the reader toward the potential utility of more recent survey data.*

- Yes, this is correct - more recent data *have been* difficult to incorporate, because no one had a method to include other data sets which used different depths.

*Line 65: Define Sc*

- defined on line 20.

*Line 66: Why does the use of pedotransfer functions make the data not comparable? Is the problem that each survey uses a different bulk density estimate approach?*

- exactly, yes. Ideally it should be measured on the same sample as the carbon fraction, otherwise we add all the additional uncertainty in the modelled (pedotransfer function) estimate of bulk density, which can be substantial but is rarely quantified. We add some text to the Discussino on this point.

*Line 68: Is estimating Sc as a function of depth an alternative to correcting with bulk density?*

- no. Bulk density is not a correction but an intrinsic part of the calculation of soil carbon stock.

*Line 78-96: For a while I thought this was already the methods section. I think the narrative here an be condensed with brief and general mention of benefits from Bayesian modeling.*

- our experience is that the choice of Bayesian statistical methods still needs to be explained in the Introduction - we are explaining the choice of method rather than the method itself. At some point in the future, these will hopefully be more widely known and understood, so that this explanation will not be needed.

*Line 97-119: condense this section and incorporate into section 1 of the introduction as 1.2 and 1.3 repeat ideas at the end of 1, but in more detail.*

*Line 154:Is it possible to refer the reader to a specific section below for more discussion of statistical issues? It could otherwise be hard to find.*

- we have added "(section 2.2 below)"

*Line 178: Do I understand correctly that survey effects are not shown in equation 8, but are included in the model?*

- correct. We have now made this explicit. We tried variants of the model with and without the survey terms (as well as various others included/excluded), and we now show the results in (new) Table 1. Adding survey as a term has small, debateable benefit (Table 1), but has quite a large cost in terms of model complexity and computation time. On balance, we think omitting it is best, hence this is the version we show in Equation 8.

*Line 243: Are data from the meta-analyses different from the other datasets? If meta-analyses are used only for comparison, this section could be removed and instead used in the discussion.*

- yes they are different and used only for comparison. However, it is useful to plot them in Fig 5, so they need to be introduced before the Results section.

*Line 251: the details on data availability may be better to add in the methods? I also got confused which data are limited – presumably not the data used in this analysis? If the availability of data has an impact on the analysis, mention that. If it does not then is it important to mention at all?*

- the issue is that for some data sets we do not know the spatial location/coordinates, and for others we cannot disclose the spatial location/coordinates, so they cannot appear on Figure 1. We have removed the confusing text in the revision.

*Line 260: instead of 'relatively infrequent' is it possible to give a percentage of complicated soil profiles?*

- We have reworded this to make it clearer. The data we have do not contain explicit information about the presence/depth/nature of soil horizons, so we cannot be more quantitative about it.

**2 Referee 2**

*In the abstract, they mention the compilation of more than 15k cores (Line 3), while the Discussion refers to more than 25k (line 381). Just make sure to double-check this number and provide a consistent value across sections.*

- both are correct: "15790 soil cores" (Line 3), and "more than 25,000 core sections" (line 381) i.e. most cores are split into several sections by depth. We changed this to "more than 25,000 individual samples (core depth sections)" to make this more obvious.

*While the authors introduce the importance of land use averages of SOC stocks, which are important for model parametrization for nationwide accountability of land use emissions (under the LULUCF/IPCC guidelines), I think that authors could better emphasize this in the discussion and propose that their results are an improvement over Bradley et al. (2005).*

- agreed - we added some text emphasising the importance at the start of the Discussion in the revision.

*Hierarchical modeling: I think that presenting the reasons for adopting a logarithm transformation of SOC with theoretical reasons and proper references is completely valuable. The explanation of the reasons for adopting a Bayesian hierarchical model was good too. However, I think that from a modeling standpoint, many of the things explained in that section were not clearly described in the Methods. For example, the definition of the priors of their analysis, the goodness-of-fit metrics for evaluating their predictions, etc. Adding this information (even as supplements) is greatly appreciated.*

We have added the explanation of the priors to the Model Development section. We did use reasonably informative priors, but the large quantity of data used here makes the analysis rather insensitive to the prior specification, so we had omitted this description previously as it does not really affect the results. Now included for completeness. We have added the goodness-of-fit metrics as a table (new Table 1) and some further description of the model selection process in the Results. In brief, we used a fairly standard range of criteria (r2, RMSE, AIC). Of these, r2 has the most intuitive interpretation, but are less straight-forward with hierarchical models, so we report the marginal and conditional r^2 values, as defined for mixed-effect models by Nakagawa et al (2017).

*It is always good to present and discuss the assumptions and notations for estimating soil carbon stocks, and this paper can serve as a reference for other works that seek to relate SOC stocks to land use change, especially because it links to UNFCCC guidelines. The rationale behind their model development is very interesting and I wonder if that can be adapted to other model structures. I mean, using the proposed whole-profile estimation by identifying the depth of the arithmetic mean, like for example, running some simple regression analysis with predicted SOC (model agnostic) and depths, etc. Of course, this depends a lot on the prediction algorithm and the way depth is treated as the predictor but seems very promising.*

- This sounds interesting, but I do not really understand what the referee means here.

*The authors mention that for the sake of brevity, treating the differences between surveys is not represented in their main model (which is already encompassed by the group-specific terms), but they provide predictions for those different sources in the results. In addition, it seems that several versions of the main model are generated for inference, and I wonder how the authors can be more transparent about all model's capacity for drawing their conclusions. They mention in the results (Line 263) that their full model (having site and location as random effects) achieved an explained variance of 90%. What about the simpler forms, e.g., when they exclude the random intercepts and slopes from group-specific terms for making Figure 4, or when another model is used to make Figure 5 (that requires the sources as random terms)? This addition is highly recommended (a table placed in supplements). This will certainly help to understand why the mean effects were significant while the prediction intervals were broad among uses and/or sources (possibly because of the low performance in some versions?).*

- addressed in the point 2nd-previous point above - we have added a table with the effect of the different terms on the variance explained as suggested, with some expanded discussion of this.

*I wonder why the y-axis in Figure 2 and Figure 3 are correctly displayed in log space (but with back-transformed labels), while Figure 4 and Figure 5 are not. I can understand it is because of the effect size visualization and the change of variable of interest, but I wonder if the 95% confidence and prediction intervals are properly displayed. I would expect higher upper intervals due to the log effect. Maybe the log effect is smoothed by bulk density when estimating the SOC stock. Would greatly appreciate a further verification of this. Similarly, I'd greatly appreciate having a better explanation of the back transformation steps in the model development.*

- Figs 2 & 3 show the highly skewed raw data, which need the log scaling to display appropriately. Figs 4 & 5 show only the means and effect sizes, so can be shown in the original untransformed units. The intervals are also be on the same untransformed scale in Figs 4 & 5. The back-transformed intervals are asymmetric, but not hugely so, and more obvious from the values than the figures. We provide these below just for clarity:

Upper prediction interval for woods, crops, grass, rough: mean +

[1] 6.280824 4.332201 5.122855 5.798258

Lower prediction interval for woods, crops, grass, rough: mean -

[1] 4.952638 3.416221 4.039831 4.572285

- The back transformation is standard procedure when working with a log-transformed response variable. Predictions are made in log space, then the exponentiated to convert back to original units. This makes the intervals asymmetric about the predicted mean.

*Figure 1: It seems that at least two data points (or cores) are misplaced in the sea. This brings me to the question about the data quality. We are not sure about which source those data points come from but might indicate that other potential errors might have happened (both for soil series and land use labeling). In fact, I went over the original Bradley paper (on the journal website it says published in 2006, but you cited 2005, must check) and there are several limitations of that dataset. The first we can spot is that the employed soil maps were made in very coarse scales (1:250,000) and this can have an enormous impact on the definition of soil series and help to explain why the authors found issues with the space-for-time substitution. Also, it sort of explains the huge prediction intervals.*

- we have removed the figure, so the comments on it no longer apply.
- On "How to cite:" from the journal website: "Bradley, R.I., Milne, R., Bell, J., Lilly, A., Jordan, C. and Higgins, A. (2005), A soil carbon and land use database for the United Kingdom. Soil Use and Management, 21: 363-369. https://doi.org/10.1079/SUM2005351"
- the 1:250,000 maps were used for spatial mapping, and were not involved in the raw soil core data we analysed here (as far as we can tell - the methods are not completely clear, and no equations are given).

*This makes me think about other ways of estimating SOC stock effect sizes across land uses, like using more advanced and very performant machine learning algorithms to make spatial maps and compare them with land use maps. There are many other sources of uncertainty involved in model building, prediction, and inference, but they could indicate the same effects with lower uncertainty in the predictions. Actually, the authors defend in the discussion that interpretability is impacted by this approach. Considering the recent advances in model interpretability (partial dependences, Shapley values, etc.), uncertainty estimation via conformal prediction, and non-parametric inference, I think it is worth investigating and comparing with their proposed method. So I recommend not opposing alternative approaches and actually stating that there is room for exploring this research problem in different ways.*

- There is certainly scope for exploration of other methods and approaches, and the machine learning field is moving fast. My general concern with machine learning is that it is less interpretable, almost by definition. Although they may give the appearance of lower uncertainty, it is hard to be sure this is not a result of over-fitting to the sample at hand, even when cross-validation is used within the available sample. So a paper comparing techniques would be worthwhile, but is not the intention here.

*Specific comments: Line 259: "(. . . ) so the variance explained by a linear trend was less". Maybe lower?*

- if we think of variance as an amount, "less" seems more appropriate.

*Line 259: I think that "complicated" profiles is not a good term, please rephrase it.*

- Changed to "soils with more complex vertical structures".

*Line 265: "The interpretation of lhe latter. . . ", amend to "the".*

- corrected

*I recommend adjusting Figure 1, especially by repositioning the legend to the bottom and keeping both map grids the same size and in different panels. Label each panel and indicate them in the caption. Please, make the notation of variables consistent across the text, figures, and tables.*

- Given comments by the other referees, we have removed the figure as unnecessary.

Reference Nakagawa, S., Johnson, P.C.D., Schielzeth, H., 2017. The coefficient of determination R2 and intra-class correlation coefficient from generalized linear mixed-effects models revisited and expanded. Journal of The Royal Society Interface 14, 20170213. https://doi.org/10.1098/rsif.2017.0213

**3 Referee 3**

*One concern is the strict applicability of the logarithmic function (soil carbon vs depth) to crop soils. These are typically ploughed to 30 cm or so, though a minor fraction may be under minimum till. This means that there is a continual dilution of the surface horizon (0-15 cm) with soil from below, and a continual deposition of carbon lower down (15-30 cm). Interestingly, we have found that plough depth in Scotland has increased over the years (Lilly and Chapman, 2015). This shift in carbon would not have been evident in the data of Jobbagy and Jackson (2000), who appeared to deal mainly with grasslands, shrublands and forests. What is also clear from the data presented here (Figure 3) is that there is quite a large data gap just in this region (on either side of the 0.25 m line). In fact, it is evident in all four land use types. What it might mean for the crops is that the surface values are less than what they would be in the absence of ploughing and that the slopes of the regression lines are less than what they would otherwise be. The net result would be a reduction of the total carbon values for these crop soils.*

- These are intersting points. One thing to clarify first is that the points in Figures 2 and 3 (now renumbered 1 and 2) represent a mean over a depth interval but are plotted with a single mid-point depth $x$ coordinate. We experimented with plotting each sample as a horizontal line representing the depth interval, but this creates more problems. So as to be able to distinguish different points, we add jitter - some random variation in the x dimension so that the points are not all over-lying on a single $x$ value. The suggestion of a gap in the data at 0.25 m depth is just an artefact of the depth intervals chosen in the surveys and plotting against the mid-point depth. To be clear, the cores are complete to whatever depth they sample, without gaps. We have added text to clarify this in the revision.

- One expected effect of ploughing would be to reduce the slope of soil carbon vs depth because of the mixing effect of the plough. This in itself should already be accounted for (we fit separate slopes and intercepts for each land use, and account for variation related to different sites and locations with the hierarchical terms), and the logarithmic decline is still likely to be reasonable. More importantly, the effect of ploughing might be to create two different layers (above and below the plough depth) with different slopes (soil carbon vs depth), so something like a "broken stick" model might be more appropriate. The practical problem is that we typically don't have enough samples in the vertical to have the resolution to see such effects. We don't see clear evidence of this in Figure 3, but perhaps only for this reason. If we knew which sites had been ploughed and to what depth, we could include this in the analysis, but the information is not generally available. We have included discussion of the above in the revised Discussion.

*A secondary consideration is the status of improved grasslands. These are also periodically ploughed but usually the time since last ploughing is unknown. It may be known to some extent where repeated samples are*

*taken, such as in the Countryside Survey. Time since being under grass will affect how close the C stock to typical crop values and how close it is to more typical 'grass' values. As argued above, there will also be an effect on the distribution of C over the Ap (ploughed) horizon.*

- as above, this is a valid point, but in practice, the information on ploughing history is not generally available. The intention is that the sample is large enough to be representative of the range found in improved grasslands, averaging over all post-ploughing states. Importantly, this is how the model used in the LULUCF inventory works, so the parameter we need to estimate is the mean for all improved grassland.

*Specific comments LL75-77 This sentence seems to be a repeat of what has already been stated in LL73-74.*

- There are two points, which we did not make clear enough:
  - the decline with depth is exponential, and so forms a linear relationship when carbon density is log-transformed;
  - the frequency distribution would be expected to be lognormal on the basis of the multiplication of bulk density and carbon fraction, and this is seen in the data e.g. Jobbagy and Jackson (2000).
- We have reworded this to be clearer in the revision.

*Figure 1 The inclusion of altitude in not really helpful, and for most of England and Wales cannot be seen. I recommend omission. Also some data points appear in the sea or in Eire!*

- agreed. We have removed Fig 1 as suggested.

*Technical corrections*

*L83 assuming*

- corrected

*L252 availability*

- corrected

*L257 Figures 2 & 3 (not S2 & S3)*

- corrected.

*L265 the*

- corrected

*L267 (and elsewhere) Use of the word 'outwith' is fine by me but may raise some eyebrows outwith Scotland.*

- Synonyms "beyond" or "outside" do not sound quite as good to me, but up to Editor's disgression (I hadn't realised this was a Scottish English word).

*L332 the assumption*

- corrected

*L356 rejects*

- corrected